# Recent Uses of Lipid Nanoparticles, Cell-Penetrating and Bioactive Peptides for the Development of Brain-Targeted Nanomedicines against Neurodegenerative Disorders

**DOI:** 10.3390/nano13233004

**Published:** 2023-11-23

**Authors:** Yu Wu, Angelina Angelova

**Affiliations:** Université Paris-Saclay, CNRS, Institut Galien Paris-Saclay, 91400 Orsay, France; yu.wu@universite-paris-saclay.fr

**Keywords:** lipid nanoparticles, liquid crystalline nanocarriers, cubosomes, cell-penetrating peptide (CPP), CPP-functionalized particles, pituitary adenylate cyclase-activating polypeptide (PACAP), neuroprotection, nanomedicine

## Abstract

The lack of effective treatments for neurodegenerative diseases (NDs) is an important current concern. Lipid nanoparticles can deliver innovative combinations of active molecules to target the various mechanisms of neurodegeneration. A significant challenge in delivering drugs to the brain for ND treatment is associated with the blood–brain barrier, which limits the effectiveness of conventional drug administration. Current strategies utilizing lipid nanoparticles and cell-penetrating peptides, characterized by various uptake mechanisms, have the potential to extend the residence time and bioavailability of encapsulated drugs. Additionally, bioactive molecules with neurotropic or neuroprotective properties can be delivered to potentially mediate the ND targeting pathways, e.g., neurotrophin deficiency, impaired lipid metabolism, mitochondrial dysfunction, endoplasmic reticulum stress, accumulation of misfolded proteins or peptide fragments, toxic protein aggregates, oxidative stress damage, and neuroinflammation. This review discusses recent advancements in lipid nanoparticles and CPPs in view of the integration of these two approaches into nanomedicine development and dual-targeted nanoparticulate systems for brain delivery in neurodegenerative disorders.

## 1. Introduction

Central nervous system (CNS)-related diseases, known as neurodegenerative diseases (NDs), mainly include Alzheimer’s disease (AD), Parkinson’s disease (PD), and Huntington’s disease (HD) [1,2]. Although about 1.5 billion people worldwide suffer from CNS-related diseases, the existing therapies for NDs primarily provide symptomatic relief but do not cure the underlying disease processes [2]. The challenges in treating NDs are not solely attributed to the involved multiple pathogenic factors. The low permeability of the blood–brain barrier (BBB) is another significant hurdle, which limits the transport of therapeutic agents into the CNS [3,4,5,6,7].

Some of the recent innovations for drug delivery to the CNS include (i) using nanotechnology to engineer nanoparticles or nanocarriers, (ii) creating receptor-mediated transport systems, (iii) investigating Trojan horse mimics for passage through the BBB, (iv) employing intranasal delivery by nanoparticles in the nasal cavity, (v) taking advantage of the intrathecal or intraventricular routes for drug transport directly into the cerebrospinal fluid, and (vi) using focused ultrasound to transiently disrupt the BBB. These targeted approaches aim to offer alternatives to traditional drug delivery, potentially enabling more effective treatments while minimizing the intrusive breach of the CNS’s protective barrier.

The purpose of this work is to briefly review CPP- and nanotechnology-based strategies, which facilitate drug transport across the BBB, the possible mechanisms of the CPP-mediated cellular uptake of drugs and other biomolecules, and their potential therapeutic uses in nanomedicine. Furthermore, the advantages of combining lipid nanoparticles and CPPs towards the higher efficacy of the nanomedicine formulations are pointed out. We highlight selected recent developments in CNS-targeted delivery systems, especially the application of liquid crystalline lipid nanoparticles and CPPs in brain delivery and the therapeutic use of biomolecules (bioactive peptides, proteins, and lipids) in the prevention and treatment of neurological diseases. Among the CPPs, we are particularly interested in PACAP (pituitary adenylate cyclase-activating polypeptide), known for its neurotrophic and neuroprotective effects.

### 1.1. Overview of the Blood–Brain Barrier Organization Controlling Drug Transport to the Brain

The BBB has been extensively studied in order to understand the possible penetration mechanisms for drugs of various molecular weights [5,6,7,8,9,10,11,12,13]. It has been demonstrated that the BBB is a highly dynamic and complex system consisting of blood vessels and cells that acts as a protective barrier between the bloodstream and the brain.

The electron micrograph of the cross section of a CNS vessel, shown in Figure 1, presents the BBB ultrastructure as being constituted by different cell types: nerve cells, supporting cells (such as astrocytes, pericytes, and microglia), and brain capillary endothelial cells (BCECs) [6,8]. Astrocytes, a prominent type of glial cells within the CNS, extend their cellular processes to envelop blood vessels, neuronal synapses, and nodes of Ranvier [11]. In addition to astrocytes, microglial cells play a crucial role in mediating immunity and neuroinflammation at this barrier site [3]. Excessive inflammation, oxidative stress, disruption of the blood–brain barrier, neuronal injury, and vascular dysfunction can all result from dysregulated microglial activation [4,5,6,7]. Neurons, glial cells (such as astrocytes), and vascular cells (including endothelial cells and pericytes) together form a neurovascular unit (Figure 1b). An essential role of the neurovascular unit is the orchestration of signals released by neurons and astrocytes to finely regulate cerebral blood flow, ensuring a sufficient supply of oxygen and nutrients to the brain [14]. Dysfunctions within the neurovascular unit are closely linked to a range of neurological disorders, including AD, stroke, and traumatic brain injury [15]. Vascular cells play a pivotal role in the BBB as they directly interface with the bloodstream. BCECs are important vascular cells in the neurovascular unit that possess tight junctions between adjacent endothelial cells and significantly restrict the paracellular flux of solutes [3]. The vascular tube is enveloped by two basement membranes (BMs), namely, the inner vascular BM and the outer parenchymal BM, also known as the perivascular glia limitans [5]. The vascular BM is an extracellular matrix secreted by endothelial cells (ECs) and pericytes (PCs), while the parenchymal BM is predominantly produced by astrocytic processes that extend towards the vasculature [8]. Those specialized endothelial cells form the walls of blood vessels in the CNS and create a polarized interface with distinct luminal and abluminal membrane compartments. This enables the precise control over the transport of substances between the blood and the brain [4]. While, on the one hand, these endothelial cells actively regulate the passage of certain substances into and out of the brain, on the other hand, they create a selectively permeable barrier to protect the brain. This duality provides opportunities for drug delivery to the CNS.

### 1.2. Strategies Using Cell-Penetrating Peptides and Lipid-Based Nanocarriers to Overcome the Challenges Imposed by the BBB

Strategies for the efficient delivery of therapeutic agents to the BBB have been actively investigated in the scope of preserving its integrity [16,17,18,19,20,21]. In the early 1990s, transferrin receptors (TfR) were found to be highly expressed at the brain capillary endothelial cells interface [13]. Subsequently, several receptor and transporter proteins, including the insulin receptor (IR), low-density lipoprotein receptor-related protein 1 (LRP1), and various amino acid transporters, were discovered at the interface of the brain capillary endothelial cells. These proteins have also been utilized in strategies for receptor-mediated drug transport for CNS targeting [7,12]. For example, Johnsen et al. investigated the interaction of immunoliposomes with brain capillary endothelial cells at the BBB and showed that targeting the transferrin receptor enhances the drug transport into the rat brain [22]. Following this finding, peptides facilitating the cellular uptake of a variety of molecules have been referred to as cell-penetrating peptides (CPPs) [23,24,25,26,27,28,29,30].

Among the CPPs, pituitary adenylate cyclase-activating polypeptide (PACAP) is a neuropeptide capable of traversing the BBB via the protein transport system-6 (PTS-6) located in the endothelium [31]. Growing evidence highlights the significance of PACAP for both its ability to access the CNS and its potent neuroprotective and neurotrophic properties [32]. PACAP has shown a remarkable neuroprotective effect and various beneficial effects against pathological states [33,34]. The exploration of bioactive peptides, like PACAP, holds great promise for crossing the BBB and represents a potential strategy for the treatment of neurodegenerative disorders [35,36,37,38,39,40,41,42].

To our knowledge, no CPPs or CPP/cargo combinations have received approval from the US Food and Drug Administration (FDA) yet, despite the increasing number of clinical trials with CPP-based delivery systems [43,44,45]. There is still limited knowledge about the immunogenicity, stability, specificity, and toxicity of CPPs. A major problem of the traditional oral administration route of hydrophilic molecules and weakly soluble drugs is their poor bioavailability. Bioactive peptides and proteins are very challenging for oral administration. A prospective strategy is to use alternative administration routes or nanoparticulate carriers to protect such therapeutic agents from enzymatic degradation [46,47,48]. In this context, the application of nanotechnology-based approaches has emerged as a promising strategy to enhance both drug targeting to the brain and the delivery of multi-drug-loaded nanocarriers (either hydrophilic or hydrophobic) [17,49,50].

Lipid-based nanocarriers, which have been utilized in the fabrication of anti-COVID-19 vaccines with high efficacy and success, are drawing considerable attention [51,52,53,54,55,56,57,58]. Such nanoparticles may facilitate the drug transport across the BBB and enable the efficient delivery of therapeutics for the treatment of neurological disorders [59,60,61,62,63]. Functionalized nanoparticles offer advantages in prolonging the residence time of therapeutic agents at the BBB and enhancing their penetration [9,20,21,22,23]. Moreover, brain delivery can be achieved using lipid nanocarriers for active targeting [64,65,66,67,68,69,70]. Active targeting consists of attaching ligands to the surface of the nanosystem or CPPs [10,12] in order to promote interaction with proteins constitutively expressed at the BBB, such as the low-density lipoprotein receptor [71,72], transferrin (Tfr) [73], and insulin receptor [49,50,74]. In addition to receptor-mediated transport, peptide-mediated delivery (CPPs) [75,76,77,78,79], and nanotechnology-based strategies [46,47,48], numerous strategies have been investigated for enhancing drug permeability across the BBB. These include chemical modification of drugs [80], osmotic disruption [81], intracerebral implantation [82], and intranasal delivery [83].

## 2. Cell-Penetrating and Blood–Brain Barrier Shuttle Peptides in Therapeutic Delivery to CNS

### 2.1. CPP Types

CPPs are amino acid sequences of variable length (in the range of 4–50 amino acids) and have a remarkable ability to cross cellular membranes and biological barriers [84,85,86]. In a cellular model, Green et al. showed the ability of a synthesized Trans-activator of Transcription (TAT) peptide, with an amino acid sequence derived from HIV-1 protein (Tat 47–57: YGRKKRRQRRR), to penetrate cell membranes [87]. TAT was the first CPP that was identified. Important advantages of CPPs are their high translocation capacity across the lipid membranes and low cytotoxicity and immunogenicity [24,25]. These peptides can facilitate the cellular uptake and delivery of cargos, like peptides, siRNA, DNA plasmids, fluorescent compounds, and nanoparticles, into cells. Various drug delivery strategies using CPPs have been explored for a range of diseases, especially for neurodegenerative disorders [26]. CPPs offer a nondestructive intracellular delivery method that does not disrupt plasma membrane structures when compared to alternative approaches in the treatment of NDs (e.g., osmotic disruption or intracerebral implantation) [27,28].

CPPs are classified according to their physicochemical properties, origin, or type of sequence. Depending on the physicochemical properties, CPPs are divided into three subgroups comprising cationic, amphipathic, and hydrophobic peptides. The cationic CPPs are the most common type of CPPs. They are highly positively charged at a physiological pH. In general, cationic CPPs contain many cationic amino acids, such as arginine (R) and lysine (K). For example, the TAT peptide is a typical cationic CPP [29]. Although the TAT amino acid sequence capable of crossing cellular barriers remains unknown, studies have suggested that charged residues, like arginine, play an important role in the efficacy of transmembrane transport [30]. Another feature is that the positive charges of CPPs may electrostatically interact with negatively charged molecules, like the phospholipids of the biological membranes [88]. Thus, the positive charge of CPPs enhances the cellular uptake efficiency. However, excessive increase in the positive charge of the CPP sequence may lead to increased cytotoxicity, as demonstrated in both in vitro and in vivo studies [89,90]. Qian et al. emphasized that arginine-rich CPPs have a serious toxicity. The authors tested the toxicity of TAT and TAT-conjugates by intravenous injection in mice [91]. The LD_50_ value of TAT was 27.244 mg·kg^−1^, which falls into the range of highly toxic chemical substances. Of note, the toxicity augments as the mass ratio of TAT in the TAT-conjugated complex augments. For comparison, a undecapeptide derivative without any arginine residue showed a much lower toxicity than TAT or oligo-arginine CPPs. This result suggests a potential connection between the toxicity of cationic CPPs and the number of arginine residues. Nonetheless, the precise toxic mechanism remains elusive.

Amphipathic CPPs have both hydrophilic and hydrophobic regions, which can fold into α-helical and β-sheet-like structures. Amphipathic CPPs can be further classified as primary amphipathic CPPs (e.g., Mannose-6-phosphate (MPG)) [92] and secondary amphipathic CPPs (e.g., penetratin) [93] depending on whether they possess an amphipathic primary sequence or acquire amphipathicity upon folding [94]. Primary amphipathic CPPs adopt a folded conformation only in the presence of cell membranes, while secondary amphipathic CPPs tend to form secondary structures in a solution prior to their interaction with lipid membranes [95].

Compared with cationic and amphipathic peptides, there are hardly many hydrophobic CPPs. These peptides typically comprise nonpolar components or include specific hydrophobic motifs crucial for penetrating the cellular membranes. They are derived from signal peptide sequences that encompass nonpolar elements, like prenylated peptides [96], pepducins [97], or staples [98,99]. Hydrophobic CPPs exhibit a low net charge and low toxicity, and their hydrophobic segments play a critical role in interacting with the lipid membranes [100]. Ongoing research aims to understand the significance and functionality of hydrophobicity for the biotherapeutic effect of CPPs [101]. Current findings suggest that the hydrophobic property of CPPs can be utilized to predict the activity of protein cargos [102].

### 2.2. Cellular Uptake Enhanced by CPPs

CPPs may enhance the cellular uptake of various cargos, including nanoparticles, proteins, drugs, and nucleic acids [50,79,80] Combining CPPs with cargos can be accomplished through various methods, including co-incubation, encapsulation, chemical conjugation (covalent bonding such as chemical linkage), non-covalent complexation, and self-assembly [103]. Since its discovery, TAT has been extensively utilized as a CPP (Table 1). TAT can traverse not only the plasma membrane but also various cellular membranes, including those of neurons, making TAT-modified NPs versatile and promising tools for intracellular delivery [98] Lee et al. have examined whether semiconductor nanowire (NW) devices and TAT-conjugated complexes (TAT-NWs) can be internalized into primary neuronal cells. First, the NWs were covalently linked to Alexa Fluor 555- labeled streptavidin (STV). Then, they were incubated in biotin-–TAT solution. Finally, TAT-NWs (5 μg) were incubated with mouse hippocampal neurons for 20 h at 37 °C. The internalization effect was examined by fluorescence microscopy (Figure 2). The result of the fluorescence microscopy imaging has demonstrated the TAT-NWs entry into neuronal cells, while bare NWs remained outside in the control group [104].

The study of Wen et al. showed that the conjugation of TAT with magnetic PLGA/Lipid nanoparticles highly enhances the therapeutic molecule delivery in a brain-derived Endothelial cells.3 (BEnd.3 cell) model by penetrating the cellular membranes [105]. Numerous studies have confirmed that TAT-conjugated nanoparticles can effectively overcome the BBB and thus can be effectively used to achieve a therapeutic dosing of the drug in the brain [106]. Furthermore, Tat-NR2B9c, a TAT-conjugated compound engineered to specifically target the N-Methyl-D-Aspartate Receptor Subunit 2B subunit of the N-methyl-D-aspartate (NMDA) receptor, exhibited promising results as a potential neuroprotective agent. It successfully concluded a Phase 2 clinical trial known as ENACT (Evaluating Neuroprotection in Aneurysm Coiling Therapy) in 2012 (ClinicalTrials.gov identifier NCT00728182). These clinical findings have stimulated further uses of CPP-conjugated drug delivery systems [107,108].

Neves-Coelho et al. investigated an anionic peptide named PepNeg (Table 1), which efficiently transports cargos, such as Green Fluorescent Protein, through the BBB without disrupting it [25]. The authors showed that PepNeg can translocate the BBB using both energy-dependent and energy-independent adsorptive-mediated transcytosis mechanisms, challenging the notion that a negatively charged membrane surface is a strict requirement for the translocation of this peptide [25].

Alizadeh et al. showed that Histone Replacement 9, a histidine-rich nona-arginine peptide (Table 1), greatly enhances the delivery of DNA into cells [62]. The Histone Replacement 9 complex led to a substantial increase in protein expression within cells compared to the control group [62]. This peptide-mediated DNA delivery system can be regarded as a promising non-covalent approach for gene transfer.

Ying et al. proposed utilizing the brain-targeting angiopep-2 peptide (Table 1) for the precise delivery of phenytoin sodium via electro-responsive hydrogel nanoparticles in antiepileptic therapy. The modified hydrogel nanoparticle system, as observed in an in vivo imaging study, demonstrated higher fluorescence intensity of Angiopep-2 in the hippocampus and brainstem regions in comparison to the control group. Additionally, the authors assessed the antiepileptic efficacy of the Angiopep-2-conjugated NP complex in an amygdala-kindling seizure model. The results confirmed the rapid, long-term, and accurate in vivo brain targeting, ultimately augmenting the antiepileptic effectiveness of phenytoin sodium through this conjugated complex [109].

Penetratin is a well-known CPP that is derived from the third helix of the Antennapedia homeodomain protein (Table 1). Alves et al. conjugated penetratin with a proapoptotic peptide Kappalactone A (KLA), which faces challenges in penetrating eukaryotic plasma membranes, resulting in minimal cytotoxicity in mammalian cells [110]. After entering the cell, the KLA peptide disrupts mitochondrial membrane integrity and initiate programmed cell death through apoptosis. To enhance the internalization of the KLA peptide, the authors initially confirmed that the KLA–Pen conjugate exhibits cytotoxicity against a panel of diverse human cancer cell lines from various tissues, including cells resistant to certain traditional chemotherapy agents [110].

The R8-based peptides contain a sequence of positively charged arginine (R) residues, which contribute to their cell-penetrating properties thanks to the interaction with the negatively charged cell surface. Ringhieri et al. demonstrated that liposomal doxorubicin, when doubly functionalized with CCK8 and R8 peptide sequences, facilitates both targeted delivery to specific cells and the efficient internalization of liposomal drugs [111]. Chen et al. illustrated through in vitro studies that the combined action of c(RGDfK) and Peptide-22 significantly enhances the uptake of liposomes by U87 cells. In vivo imaging subsequently confirmed that c(RGDfK)/Pep-22-LP liposomes exhibited a greater distribution within brain tumors compared to liposomes modified with a single ligand [112].

CPPs have also found applications in the delivery of DNA vaccines. Saleh et al. used a gene delivery system based on MPG, which forms stable non-covalent NP complexes with nucleic acids, suitable for both the in vitro and in vivo delivery of HPV16 E7 DNA. The latter encodes for a model antigen of human papillomavirus (HPV) type 16. The results convincingly demonstrated that MPG effectively condensed plasmid DNA into stable nanoparticles, featuring an average size ranging from 180 to 210 nm and a positively charged surface. Furthermore, the transfection efficiency of MPG-based nanoparticles was found to be like that of polyethyleneimine (PEI). The robust protein expression detected via Western blotting and flow cytometry underscores the promising potential of MPG-based nanoparticles as a potent delivery system in DNA vaccine formulations [113]. A synthetic CPP peptide C105Y derived from alpha1-antitrypsin 359–374 (Table 1) was reported to exhibit a high propensity for rapid internalization in living cells (by a clathrin- and caveolin-independent pathway) and enhancing gene transfer to the nucleus and the subsequent gene expression [114]. Membrane translocation and nuclear localization were suggested to be energy-independent, whereas the C105Y peptide traffic to the nucleolus occurred in an energy-dependent fashion.

**Table 1 nanomaterials-13-03004-t001:** Amino acid sequences of CPPs used in peptide-assisted drug delivery and examples of blood–brain barrier shuttle peptides referred to in this review.

Peptide Name/Type	Amino Acid Sequence	Reference
TAT (Trans-Activating Transcriptor)	CGRKKRRQRRRK	[28,29,91]
PACAP-38	HSDGIFTDSYSRYRKQMAVKKYLAAVLGKRYKQRVKNK	[115,116]
PepNeg	SGTQEEY	[25]
HR9	CH-HHHHRRRRRRRRRHHHHHC	[62]
Angiopep-2	TFFYGGSRGKRNNFKTEEY	[109]
Penetratin	RQIKIWFQNRRMKWKK	[110]
HAI	H-HAIYPRH-NH2	[117]
R8 peptide	YARAAARQARA	[111]
Peptide-22	NH2–NH2–CGGGPKKKRKVGG–COOH	[112]
MPG	GALFLGFLGAAGSTMGAWSQPKKKRKV	[113]
C105Y	CSIPPEVKFNKPFVYLI	[114]

### 2.3. Blood–Brain Barrier Shuttle Peptides

The blood–brain barrier shuttle peptides are a subclass of CPPs, which allow varieties of cargoes to cross the BBB, e.g., conjugated small molecules, proteins, nanoparticles, or genetic material. Small peptides with the ability to pass the BBB, including enkephalin, fragments of transferrin, and insulin-like growth factors, were identified by Pardridge in 1986. This finding prompted suggestions to use these peptides as delivery systems for brain targeting [12]. Then, in vitro and in vivo studies have demonstrated the brain-targeting properties of some peptides. PepH3, for instance, has demonstrated its ability to cross the BBB in both in vivo biodistribution and in vitro experiments [118]. Small peptides are current-ly being explored as potential strategies for transporting substances from the blood to the brain. Among these, Angiopep-2 has drawn attention for its effective transport of therapeutic payloads, like siRNA [119], enzymes [120], and drugs [121], across the BBB. Demeule et al. have investigated a family of peptides named Angiopeps in an in vitro model of the BBB and in situ brain perfusion. The results have suggested that the abil-ity of Angiopep-2 to cross the BBB is mediated by its interaction with the LRP1 recep-tor [118]. Furthermore, Drappatz et al. have reported a Phase I study of a formulation (GRN1005) consisting of Angiopep-2 and paclitaxel for the treatment of glioma.They indicated that GRN1005 exhibits a comparable toxicity profile to paclitaxel in patients but significantly increased the accumulation of paclitaxel within the tumor. This illustrated the potential of Angiopep-2 as a transporter of therapeutic agents across the BBB with minimal toxicity [45].

Nevertheless, there are certain difficulties in using blood–brain barrier shuttle peptides. Making sure these peptides target the correct sections of the brain without impacting other areas, which may have unexpected effects, is a major concern. Furthermore, there are restrictions on the size and kind of materials that these peptides can transport across the BBB. Concerns have also been raised over the duration of these peptides’ presence in the body and the possibility of adverse effects. These problems indicate that much research needs to be done before these peptides are widely applied in nanomedicine.

While there are still limitations associated with the application of blood–brain barrier shuttle peptides, structural modifications can improve their applications. The noncanonical anionic peptide PepNeg, which was designed using PepH3 as a template, can efficiently transports cargo (up to 27 kDa) through the BBB model without disrupting the barrier when compared to PepH3 [25]. Li et al. synthesized a CPP called Trans-activator of Transcription-Neurotensin, which inhibits the nuclear translocation of annexin A1 (ANXA1), reduces caspase-3 apoptosis pathway activation, and enhances the survival of oxygen-glucose-deprived hippocampal neurons in vitro. Using unilateral intracerebroventricular injection, the efficient delivery of the Trans-activator of Transcription-Neurotensin peptide to the ischemic hippocampus and cortex has been demonstrated in the animal model of brain ischemia [122]. A hyaluronic-acid-modified peptide, which has an affinity to the human transferrin receptor (TfR) in brain capillaries, was able to provide transporter-mediated delivery across the BBB. Arranz-Gibert et al. showed that the hyaluronic-acid-modified peptide was able to deliver AuNPs to the brains of rats [117]. Prades et al. demonstrated that the peptide modified by a retro-enantio approach has a low toxicity, higher stability against proteases, and improved capacity of transporting drugs into brain [43]. Furthermore, the study of Oba et al. indicated that the addition of cyclic dAAs and stapled peptide into an R9 peptide sequence (RRRRRRRRR) increases the cell penetrating abilities due to the presence of dAAs, which stabilize the helical secondary structure of CPPs [123].

### 2.4. Peptide Internalization Mechanisms in Relation to the Peptide-Mediated Passage of the BBB

The precise internalization mechanism of CPPs remains a question as it depends on various parameters, such as the peptide sequence, peptide concentration, the therapeutic molecule being conjugated, and the lipid constituents of the cellular membranes. Therefore, it is crucial to thoroughly examine the patterns of CPPs’ entry into cells towards a comprehensive assessment of their safety and effectiveness [124]. To date, two major membrane-crossing routes have been widely accepted: (i) direct translocation and (ii) endocytosis (Figure 3). Endocytosis is an energy-requiring process of active molecular transport. The direct translocation of CPPs across biological barriers is an energy-independent mechanism that can be evaluated under specific experimental conditions, such as low temperature, energy depletion, and the application of endocytic inhibitors [125]. This mechanism mainly includes three different models, the transient toroidal pore formation (e.g., MPG [126]), inverted micelle (e.g., TAT [127]), and the ‘carpet’ model (magainin 2 [128,129]). There are two models for pore formation: the barrel stave model and the toroidal model. In the barrel stave model, the CPP forms a barrel with the hydrophobic surface near the lipids and hydrophilic interface inside. In the toroidal model, the lipids bend, keeping the CPPs close to the surface, and both CPPs and lipids create a pore [125]. The inverted micelle model was introduced earlier as a potential mechanism for the direct penetration of penetratin, primarily relying on interactions between hydrophobic components, such as tryptophan, and the hydrophobic region within the membrane [130]. The ‘carpet’ model may alternatively be referred to as the ‘membrane-thinning effect’ because it reflects the interaction between positively charged CPPs and negatively charged membranes. When CPPs aggregate on the membrane surface, they induce a decrease in the local surface tension, enabling them to intercalate into the membrane [131].

Crossing the BBB involves multiple transport mechanisms that offer the possibility of drug delivery to the brain [131,132]. (i) Passive crossing, including passive diffusion and adsorption-mediated transcytosis. There are non-specific transport mechanisms for passive crossing. (ii) Active crossing, involving specific transporters for small molecules (such as glucose and amino acids). Certain proteins, such as transferrin or insulin, bind to specific receptors.

The cellular uptake of arginine-rich peptides has been suggested to occur through endocytosis and macropinocytosis mechanisms (Figure 3) [124]. The cellular uptake of PACAP involves direct translocation and endocytosis, i.e., this CPP may cross the lipid membrane by a receptor-independent mechanism. As a bioactive peptide, PACAP plays a pivotal role in a wide range of physiological processes and signaling pathways, particularly in neural function. Its distribution has been characterized in both the CNS and in peripheral organs [133]. PACAP employs also a receptor-dependent internalization mechanism through three primary receptors: the PAC1 receptor as well as the vasoactive intestinal peptide receptor 1 and VPAC2 receptors [134]. Over ten years, receptor-mediated endocytosis was believed to be the potential mechanism for PACAP to translocate into the intracellular compartments. However, Doan et al. proposed a receptor-independent cellular uptake of PACAP [135]. Understanding the internalization mechanism of PACAP is essential for elucidating the process of signal transduction from the cell periphery to the nucleus.

Other CPPs, e.g., PepNeg, can also use two routes to penetrate the BBB. Whereas the direct translocation occurs with some low-molecular weight components, the internalization mechanism depends on the CPP concentration [136]. Cesbron et al. revealed the internalization of HA2 fusion peptide-functionalized gold nanoparticles in HeLa cells by transmission electron microscopy (TEM) (Figure 4). The obtained results indicated that the conjugated nanoparticles (NPs) are capable of entry into the cells. Moreover, grafted PEG chains slightly augmented the NP density in the endosome [137].

## 3. Nanoparticle-Mediated Drug Delivery

### 3.1. Lipid Nanoparticle Types

Nanotechnology-based delivery systems are the most common delivery systems for the efficient transport of therapeutic agents [138]. Nanoparticles, extensively explored in a variety of disease treatments and diagnoses, offer a wide range of possibilities for drug delivery [139,140,141]. Lipid nanoparticles, which comprise colloidal dispersions, have been vastly reported as safe and efficient nanocarriers [142,143,144,145,146]. Depending on their physico-chemical properties, lipid nanoparticles can be divided into several categories, such as liposomes, solid lipid nanoparticles, nanostructured lipid carriers, nanoemulsions, niosomes, and a cubosome-type of liquid crystalline nanoparticles (Figure 5). The diverse types of lipid nanoparticles have special features that may impact brain drug delivery, particularly in terms of drug release and stability. Choosing the right lipid nanoparticles is crucial for the efficient drug delivery to the brain. Comprehending the various lipid nanoparticle types and their distinct characteristics should facilitate the optimization of the drug delivery process to the complexity of the brain environment, towards the treatment of neurological conditions.

Liposomes have been extensively invested as delivery systems for brain targeting [50]. Liposomes are composed of a spherical aqueous core encompassed by a bilayer of phospholipids. In general, liposomes cannot overcome the BBB without targeting ligands. However, they may augment the drug concentration in the brain by prolonging the circulation time. Another advantage of liposomes is that they can encapsulate both hydrophilic and hydrophobic compounds [147]. Liposomes conjugated with a targeted peptide may essentially improve the drug efficacy. The study of dos Santos Rodrigues et al. reported a formulation of liposomes conjugated with CPPs and transferrin (Tf), which exhibited a higher ability to cross the BBB. The results of the performed in vivo studies confirmed that TAT-Tf liposomes transport therapeutic DNA into the mice brain [142]. Therefore, the CPP-functionalized liposome system holds considerable promise for delivering therapeutic agents to the brain.

Solid lipid nanoparticles (SLNs) contain a solid lipid core at room and body temperature. SLNs can encapsulate a variety of therapeutic molecules, protect them from reticuloendothelial system clearance, and reduce their toxicity [148]. However, due to the solidification and subsequent crystallization of the lipid phase, the SLNs may have a serious problem of stability. Nanostructured lipid carriers (NLCs) are colloidal systems formed by binary mixtures of lipids. NLCs can avoid the recrystallisation and phase separation of the encapsulated active molecules. NLCs have enhanced stability and offer controlled release, crucial for maintaining drug levels in the brain over time. Thus, NLCs are considered to be more stable than SLNs [144,145,146].

Nanoemulsions are oil-in-water (O/W) or water-in-oil (W/O) dispersions of two immiscible liquids stabilized by appropriate surfactants. They comprise small droplets with sizes from 20 to 400 nm. The advantages of nanoemulsions compared to conventional emulsions are their higher stability, vast surface area, and rapid absorption. The administration routes for nanoemulsions include parenteral, oral, topical, and intranasal drug delivery systems [149,150].

Cubosomes are lipidic nanoparticles with an internal cubic structure of bicontinuous cubic or micellar cubic types. They can be formed by the self-assembly of lipid mixtures involving nonlamellar lipids [57,66]. The bicontinuous lipid cubic phase encompasses a lipid bilayer with a periodic minimal surface intertwined with a three-dimensional network of water nanochannels. The advantages of cubosomes are biocompatibility, possibility for the co-encapsulation of hydrophilic and hydrophobic compounds, stability in biological milieu, and eased internalization by the cells [151,152]. Cubosome-type liquid crystalline nanoparticles, due to their unique inner self-assembled cubic membrane structure, offer increased drug encapsulation rates and controlled drug release profiles. Sustained drug release from nanocarriers appears to be of crucial importance for brain drug delivery. Rakotoarisoa et al. showed the capacity of cubosomal nanoformulations for the co-delivery of curcumin and catalase to human neuroblastoma cells (SH-SY5Y) studied as a cellular model of neurodegeneration. The results indicated that curcumin-containing cubosomes can maintain or increase the enzymatic catalase activity. Such nanoformulations exhibited neuroprotective properties [153]. The uptake mechanism of cubosomal NPs can be impacted by their surface architecture and coating. Deshpande et al. established different uptake mechanisms for poly-ε-lysine (PεL)-coated cubosomes and uncoated blank cubosomes [75].

The delivery of drugs to CNS is impacted by the nanocarrier type [154,155,156,157]. Table 2 presents examples demonstrating the significance of the discussed LNP types for the resulting drug delivery to the brain or spinal cord.

### 3.2. Targeting and Internalization Capacities of Nanocarriers

To develop nanoparticle (NP)-based carriers for efficient drug delivery across the BBB, several questions must be addressed. One needs to determine whether NPs can successfully pass the BBB, and if so, elucidate the mechanisms by which NPs achieve this penetration. Nanoparticles can traverse the BBB via several mechanisms [163]. Small lipophilic molecules (<400 Da) diffuse passively through the endothelial cell layer [164]. For relatively larger biomolecular ligands, the process involves receptor-mediated transcytosis. Specific surface-exposed ligands bind to endothelial cell receptors, prompting the nanoparticles’ internalization and transport across the barrier. Additionally, adsorption-mediated transcytosis may take place when nanoparticles interact with cell membrane components due to their specific charge or surface properties. Moreover, certain nanoparticles or drug delivery systems can induce temporary BBB disruption to facilitate their passage. The capability of NPs to traverse the BBB depends on various parameters, including size, shape, ligand density and binding affinity, and surface charge. By varying these parameters, the BBB-crossing ability and internalization mechanism of NPs may be altered [165]. Liu et al. established that the size and shape of TiO_2_ particles have an impact on their ability to permeabilize the BBB. Specifically, small spherical TiO_2_ NPs exhibited a higher efficiency in permeabilizing the BBB compared to larger, rod-like particles [166]. The primary mechanism for NPs to cross the BBB involves transcytosis through the endothelial cell layer, which provides options for enhancing BBB permeability for drug delivery to targeted cells [167].

To improve the targeting capacity, the surface of magnetic nanoparticles was modified with moieties, such as poly (D,L-lactide-co-glycolide) (PLGA), and CPPs. The modified NPs are referred to as the third generation of “targeted NPs” [168]. As mentioned above, the targeting often includes the TfR receptor and other peptides, which are highly expressed on the BBB endothelial cells [142].The intravenous administration of doxorubicin-loaded nanoparticles markedly extended the survival times of rats with a single brain tumor. Fluorescence images of tumor-bearing mouse brain sections revealed significant nanoparticle and doxorubicin accumulation around large tumors [169]. Householder et al. presented direct biodistribution evidence, demonstrating that 100 nm nanoparticles rapidly distribute within the subarachnoid space of the brain and spinal column following intrathecal administration in healthy mice [170]. c et al. demonstrated that chitosan nanoparticles (NPs) loaded with the neuroprotective peptide basic fibroblast growth factor (bFGF) and conjugated with the anti-transferrin receptor-1 monoclonal antibody (TfRMAb) rapidly penetrate the BBB following systemic administration to mice through transferrin receptor-mediated transcytosis. This formulation provided enhanced neuroprotection against focal cerebral ischemia. The targeted delivery not only conferred neuroprotection but also substantially reduced the risk of systemic side effects by decreasing the administered bFGF dosage by approximately 300-fold [171].

Lipids constitute an alternative choice to polymer nanocarrier systems for drug delivery to brain (Figure 6), as only a limited number of polymers have received regulatory approval for clinical use to date [54,56,128], Lipid nanoparticles (LNPs) are particularly suitable for the formulation of lipophilic pharmaceuticals and have been extensively investigated as carriers for a large number of therapeutic agents against NDs (e.g., curcumin [172], ropinirole (RP) [173], and quercetin [174]). In addition, lipid nanoparticles have advantages, such as biocompatibility and biodegradability, contributing to their overall safety profile [175]. They can incorporate both hydrophilic and hydrophobic drugs (as single compounds or in a combination). Moreover, LNPs are easily scalable for manufacturing, possess the ability to encapsulate and protect biotherapeutics, exhibit good long-term stability, offer prolonged drug release effects, have small sizes that facilitate cellular internalization, allow for surface modification to promote drug targeting, and facilitate drug transport across the BBB [103,104,105,106,107].

While lipid nanoparticles show promise for delivering drugs to the brain due to the ability of drugs to cross the BBB, their efficacy is hindered by several limitations. One of the constraints is the induction of immune responses, leading to clearance by the body or causing adverse reactions [176,177]. Targeting specific brain regions remains a hurdle because the complex biological environment of the brain may impede the efficient transport and release of therapeutic payloads from the nanoparticles [178]. Therefore, although lipid nanoparticles provide pathways for brain drug delivery, the existing drawbacks need to be addressed for the successful application of LNPs in treating neurological disorders.

## 4. Examples of Nanomedicine Development for Neurodegenerative Diseases

### 4.1. Bioactive Peptides for Neuroprotection and Neurorepair

Bioactive peptides are amino acid sequences obtained by either chemical synthesis or via enzymatic hydrolysis of proteins (e.g., protein hydrolysates from food resources). Such peptides can be applied in treatment approaches against various human diseases [75]. Many publications have suggested that some bioactive peptides with neuroprotective effects (e.g., PACAP) can exert a potential therapeutic role in the prevention and treatment of neurodegenerative diseases. PACAP (Table 1) was originally isolated from ovine hypothalamic extracts in view of its capacity to stimulate cyclic adenosine monophosphate formation in rat anterior pituitary cells [115]. Over the past three decades, PACAP has been shown to exert neurotrophic and neuroprotective effects in both in vitro and in vivo models of various neuropathologies [116]. While PACAP possesses cell-penetrating capacities, its neuroprotective potential has gained more attention, positioning it as a promising therapeutic agent rather than a conventional CPP. PACAP has been proven to activate CREB (cAMP response element-binding protein), an important transcription factor involved in various cellular processes. The binding of PACAP to its receptors triggers signaling cascades (ERK, AKT, or PLC pathways) that lead to the activation of CREB (Figure 7) [179]. In AD pathology, low PACAP levels have been observed in cerebrospinal fluid (CSF) samples and the brain tissue of human patients, which correlated with cognitive decline during mild cognitive impairment and dementia stages [180]. These data confirm the significant role of this bioactive peptide for maintaining the neurons’ function.

PACAP treatment has shown effects in slowing down AD progression by protecting neurons against the toxicity of Aβ-amyloid-42 oligomers [181]. This protection likely occurs by enhancing mitochondrial function [181]. Chen et al. identified several genes that were upregulated following middle cerebral artery occlusion (MCAO) in both wild-type and PACAP-deficient mice and subsequently suppressed by PACAP treatment after MCAO. This transcriptional pattern aligns with the concept that these genes play a role in injury response, as exogenous PACAP treatment is associated with improved neurological outcomes following stroke [182]. Additionally, PACAP binding has been observed in the substantia nigra and ventral tegmental area, both of which are crucially involved in the pathomechanism of Parkinson’s disease [183]. The neuroprotective properties of PACAP have also been demonstrated in a rat model of PD, where peptide administration prevented the degeneration of nigral dopaminergic neurons, slowed down cognitive decline, and ameliorated behavioral deficits by regulating dopamine levels and PD protein 7 (PARK7) [184]. While there is a substantial body of evidence highlighting the pivotal role of PACAP and its therapeutic potential against NDs, certain challenges must be overcome before translating it as a pharmaceutical product. Delivering neurotherapeutic molecules, like PACAP, to CNS poses some challenges. The presence of the BBB and some efflux mechanisms in the brain make the delivery problematic for neurotherapeutic agents of peptide or protein types.

Zhang et al. demonstrated that round scad hydrolysates (RSH), containing two antioxidant peptides, HDHPVC and HEKVC, exhibit robust neuroprotective properties, including a range of activities such as antioxidation, anticoagulation, and blood pressure reduction. They have also provided further evidence confirming the protective effects of RSH in mitigating memory deficits induced by sleep deprivation in rats [185].

Leo et al. recently tested the neuroprotective effects of several peptide fractions from *S. hispanica* on HMC3 cells. F-1–3 kDa from *S. hispanica* significantly increased HMC3 cells’ viability after tert-butyl hydroperoxide (TBHP) damage. This result proved the neuroprotective effect of the F-1–3 kDa peptide. The intracellular ROS, NO production, H_2_O_2_ production, and TNF-α were examined in order to understand the neuroprotective mechanism of the F-1–3 kDa peptide. The production of ROS, NO, H_2_O_2_, and TNF-α were all decreased after the incubation of the F-1–3 kDa peptide. Thus, the neuroprotective effect of the F-1–3 kDa peptide could contribute to anti-inflammatory and antioxidant mechanisms [154].

In the case of Alzheimer’s disease, the use of peptide inhibitors targeting amyloid β can mitigate the accumulation of this protein, a hallmark of AD pathology [85]. Amyloid β (Aβ) peptides are fragments derived from a larger protein known as amyloid precursor protein (APP), and they have the propensity to aggregate into toxic oligomers and fibrils. Aβ peptide inhibitors are designed to interfere with or regulate the aggregation process of Aβ peptides, with the objective of either preventing or slowing down the formation of amyloid plaques [186]. Consequently, amyloid β peptide inhibitors have been the focus of research in AD treatment for the past two decades. These inhibitors may be tailored to address the various stages of Aβ aggregation, encompassing monomers, oligomers, and fibrils, and their development holds great promise for the advancement of therapeutics for Alzheimer’s disease (as illustrated in Figure 8) [187].

### 4.2. Lipid Nanoparticles for Neuroprotection and Neurorepair

Lipid nanoparticles (LNPs) are biocompatible and biodegradable and help to alleviate the side effects linked to conventional drug administration [52,143,165,188,189,190]. In addition to their safety, LNPs aid in preventing the degradation of encapsulated bioactive molecules [144,145,146,147,149,150,151]. Dual drug-loaded cubosome-type nanocarriers, fabricated by self-assembly, can display multicompartment liquid crystalline organization involving nanodomains (Figure 9). The regulation of endoplasmic reticulum (ER) stress by dual-loaded liquid crystalline LNPs has been suggested as a therapeutic option for the prevention of neurodegeneration [172].

To overcome the problems of peptide and protein delivery, lipid-based nanocarriers have been widely investigated for nanomedicine development due to the ability to protect the encapsulated therapeutic components, prolong the circulation time, and enhance the targeting effect by modifying the LNP surface. Fukuta et al. demonstrated the efficacity of liposomal drug delivery systems for ischemic stroke therapy. The prepared PEG liposomes were intravenously injected in the rat model. The DiL fluorescence was detected in slices of the ischemic region using an imaging system. The results indicated that PEG liposomes can accumulate in the ischemic region. Moreover, the accumulation in the ischemic region was enhanced in comparison to that in the non-ischemic region, which implied that the PEG liposomes can penetrate through the EPR effect (enhanced permeability and retention effect) in the diseased region [191]. Gajbhiye et al. reported the use of PEGylated nanocarriers as an efficient targeted drug delivery to the brain. The PEG chains prolonged the circulation time for the drug to entry the brain [49]. Other examples using dual-modified liposomes were presented in the study of Rodrigues et al., which showed that liposomes conjugated with a CPP and Tf can cross the BBB after a single intravenous administration. In addition, the results of in vivo studies confirmed that the TAT-Tf liposomes facilitate the transport and the entry of therapeutic DNA into the mouse brain. Therefore, this system appears to be very promising for brain-targeted gene delivery [67].

Although numerous peptides and proteins have demonstrated their neuroprotective properties in cellular and animal models, translating these findings to clinical applications poses a significant challenge. The human body is an intricately interconnected system, making the precise delivery of therapeutic agents to the brain an important task. LNPs are suitable for neuroprotection and neurorepair thanks to their advantages including the encapsulation of hydrophobic drugs and increasing their bioavailability after delivery to the target tissue. For example, andrographolide (AG) has a neuroprotective effect but poor solubility and low bioavailability. Graverini et al. showed that solid lipid nanoparticles (SLNs) can deliver AG into the brain. SLNs improved the release profile and drug transport to the BBB site. These results were confirmed by evaluations using an in vitro cellular model, a BBB model, and in vivo experiments in rats [143]. Zhu et al. proved the effect of gelatin nanostructured lipid carriers (GNLs) with encapsulated neural growth factor (NGF) to reduce the neuronal deficits in a spinal cord injury (SCI) model. The authors studied neuronal survival, behavioral recovery, and measured the effect of SCI recovery by hematoxylin–eosin staining. The results suggested that the administration of NGF-GNL leads to neuroprotective outcome by the regulation of endoplasmic reticulum ER stress [162]. An example of a neuroprotective effect of PACAP on the CNS was shown in the study by Ho et al., who investigated the in vivo efficacy of controlled release of PACAP from a nanoparticle–hydrogel composite [192]. To achieve the therapeutic effect, the created controlled release system was employed for the topical delivery of PACAP to the brain of mice with stroke injuries over a 10-day period of treatment.

### 4.3. Neurotherapeutic Delivery for the Treatment of Alzheimer’s Disease

The exploration of AD treatment drugs is still ongoing. AD stands as the most prevalent progressive neurodegenerative condition, posing one of the most significant challenges for healthcare over the years. Its symptoms encompass memory loss and cognitive decline, both of which profoundly affect a patient’s daily life. The number of AD cases continues to rise annually, particularly in developed countries. According to the World Health Organization (WHO), projections suggest that, by 2050, the global count of patients may reach as high as 152 million [193]. AD is a complex, multifactorial condition [194]. The overexpression and accumulation of amyloid β (Aβ) aggregates have been widely considered as a prominent factor in the disease’s pathogenesis. Consequently, a substantial majority of clinical trials and research efforts have been focused on developing drugs and interventions that target Aβ to slow down the progression of AD. Several drug development programs are currently in phases 2 and 3. Because many trials are stopped early on, it remains unclear whether longer term treatments would have exerted beneficial effects [194,195].

#### 4.3.1. Peptide-Based Strategies

Peptide-based strategies for AD have primarily centered on reducing the accumulation of amyloid β plaques. The peptide-based formulation CH-3 is a bovine casein hydrolysate produced by three enzymes. Akio et al. assessed its ACE-inhibitory activity and antihypertensive effects, demonstrating significant antihypertensive effects when compared to other hydrolysates [196]. Min et al. demonstrated that orally administering CH-3 peptides can improve cognitive function in an AD mouse model. In that study, the AD animal model was created by the intracerebroventricular injection of Aβ_1-42_ in mice, leading to cognitive impairment as confirmed by Morris water maze testing. Following the oral administration of CH-3, cognitive function was restored to levels comparable to the control group. Furthermore, a tripeptide named MKP, distinct from CH-3, was also examined in the AD mouse model. MKP exhibited the capability to penetrate the BBB and effectively mitigated Aβ_1-42_-induced cognitive impairment [197].

In addition to those neuroprotective peptides, another peptide-based strategy targeting AD involves Aβ or Tau peptide vaccines, composed of various fragments of Aβ or Tau peptides. These vaccines function by training the human immune system to recognize and eliminate the harmful Aβ deposits (or Tau protein tangles) present in the brains of Alzheimer’s patients. One such vaccine candidate is ACI-35, designed as a liposome-based vaccine containing 16 copies of a synthetic Tau fragment (Tau393–408) with phosphorylation at residues S396 and S404. These Tau phospho-peptides are modified to include two palmitic acid chains at each end, allowing them to assemble into liposomes. At present, these promising vaccines are undergoing preclinical and clinical studies to evaluate their safety and efficacy [198].

#### 4.3.2. Protein-Based Strategies

Proteins play a central role in maintaining the normal function of the CNS. Neurotrophins (NTs) are a group of proteins with neuroprotective effect and potential therapeutic effect against AD, PD, and Huntington’s disease. Neurotrophins can prevent or reduce neuronal degeneration through their neurotrophic action on the specific neuronal populations. The mammalian neurotrophin family includes nerve growth factor (NGF), brain-derived neurotrophic factor (BDNF), neurotrophin-3 (NT-3), and andneurotrophin-4/5 (NT-4/5) [155,156]. Glial-cell-line-derived neurotrophic factor (GDNF) is an important protein, which maintains the survival of midbrain dopaminergic neurons [155]. The dysfunction of GDNF is related to many neurodegeneration diseases. A study reports that the level of GDNF decreased in the plasma of AD patients [157]. Thus, the administration of GDNF may be a potential treatment for some CNS-related diseases. However, GDNF does not cross the BBB. Dietz et al. showed the efficacity of TAT to deliver GDNF across the BBB in a mouse model of PD. The performed immunohistochemical analysis confirmed that Tat-GDNF can reach the brain areas after systemic administration [156]. These protein-based strategies presented low toxicity in preclinical investigations, but further formulation and safety studies will be required before eventual clinical trials.

### 4.4. Combination of Cell-Penetrating Peptides with Nanoparticles for the Treatment of Neurodegenerative Diseases

Neurodegenerative diseases, such as AD and PD, present significant challenges for the targeted delivery of therapeutic agents to the brain [52,57]. The combination of CPPs and nanoparticles offers several advantages, such as increased cellular uptake and the targeted delivery of specific cells or disease sites, which minimizes side effects and improves therapeutic outcomes. Several studies have emphasized the beneficial effects of CPP-NPs conjugates to treat a variety of diseases (e.g., cancer), CNS disorders, topical impairments, as well as imaging probes [199]. Ahlschwede et al. showed that the conjugation of a cationic CPP (K16ApoE) with NPs not only significantly improves the transport of therapeutic molecules into the brain but also targets a specific peptide (Aβ_40_) involved in the AD pathology. This result indicated that CPP-NPs conjugates can be a promising delivery system for AD treatment [200]. Zhang et al. reported on a dual-targeting drug delivery system, which transports a siRNA therapeutic agent for AD treatment. This formulation can knock down the beta-amyloid-converting enzyme 1 (BACE1) to reduce Aβ formation. Furthermore, the dual-modified NPs showed low toxicity in a cellular model [201]. Wen et al. showed the targeted efficacity of a multifunctional targeted drug delivery system, TAT-conjugated magnetic poly (D, L-lactide-co-glycolide) (PLGA)/lipid NPs (MPLs) (Figure 10). The preparation process is illustrated in Figure 10a, whereas strong fluorescence from TAT-MPLs was observed using laser confocal scanning microscopy (Figure 10b). This finding indicates that TAT-MPLs are capable of efficiently delivering the fluorescent conjugates (FITC and QDs) into bEnd.3 cells [105].

While CPP-functionalized nanoparticles are still being investigated, there are obstacles for their widespread use in the treatment of NDs. Even though CPP-nanoparticle combinations facilitate intracellular drug delivery, they face difficulties with stability, possible cytotoxicity, off-target effects, and poor selectivity in different biological settings. Concerns about immunogenicity and body clearance, arising from the interplay between CPPs and NPs, may compromise the efficiency of cargo delivery, which would ultimately reduce therapeutic efficacy. Strategies such as the ‘nose-to-brain’ delivery enable more rapid drug delivery to brain compared to traditional delivery routes and provide alternatives for improving the treatment of neurodegenerative diseases and related CNS disorders.

## 5. Perspective Applications of the Nose-to-Brain CPP-Mediated Delivery of Bioactive Molecules in Nanomedicine

The treatment of neuronal diseases presents a significant challenge owing to the BBB, which limits the delivery of most biomolecules in therapeutic quantities to the CNS. In addition to the challenge posed by the BBB, many neuroprotective agents, such as PACAP, have short half-lives [115]. Specifically, when PACAP is administered intravenously (IV), it exhibits a remarkably brief half-life of less than 5 min due to its rapid degradation by dipeptidylpeptidase IV (DPP IV) (an exopeptidase responsible for cleaving X-proline or X-alanine dipeptides from the N-terminals of the polypeptides) [115]. In addition to IV administration, direct injection into the brain is not considered a readily available translational option due to its highly invasive nature and impractical clinical application. An alternative and effective strategy for drug delivery to the brain can be achieved with noninvasive nose-to-brain drug delivery systems. The intranasal route has received considerable attention for its ability to deliver drugs directly to the brain without passage through the systemic circulation [202]. While the exact pathway from the nose to the brain remains somewhat unclear, Pardeshi et al. suggested that drugs traverse to the brain through both the olfactory nerve pathway and the trigeminal nerve pathway [203].

Research into the intranasal drug delivery systems of small-molecular-weight drugs, peptides, and proteins experienced substantial growth over the past decade [161,204]. For instance, Boche et al. evaluated the efficacy of a nanoemulsion formulation in delivering quetiapine fumarate (QTP) to the brain. The QTP nanoemulsion, when administered intranasally, yielded a higher peak concentration in the brain (0.48 ± 0.16 μg/mL) compared to the formulation administered intravenously (0.25 ± 0.44 μg/mL). This indicated that intranasal nanoemulsion administration is an efficient approach to deliver the therapeutic molecule to the brain [205]. Brown et al. conducted research on the antidepressant effects of a TAT-conjugated D1–D2 interfering peptide after intranasal administration. TAT-D1–D2 interfering peptide was administered to a rat model of depression using a pressurized olfactory delivery (POD) device. The outcomes of the study were assessed by the forced swimming test, immunofluorescence, and confocal microscopy imaging. The results of the forced swimming test showed that the immobility counts for the TAT-D1–D2 interfering peptide and Imipramine significantly decreased compared to the Saline group and the TAT group, indicating the antidepressive effect of the TAT-D1–D2 interfering peptide. Additionally, immunofluorescent staining of prefrontal cortex slices from treated rats confirmed the capability of TAT-D1–D2 interfering peptide to enter the CNS. However, the mechanism by which TAT-D1–D2 interfering peptide enters the CNS remains a question [206]. To enhance nose-to-brain drug delivery, various approaches, such as nanosystems with chitosan-surface modification, PEG-surface modification, lipid surface modification, and peptide surface modification, have been explored [207]. There are several ongoing clinical trials exploring intranasal therapeutics against neurodegenerative diseases [208].

## 6. Conclusions

In recent years, there has been a significant interest in the use of lipid nanoparticles and CPPs for achieving a more efficient targeted delivery for the treatment of CNS disorders. In this review, we summarized recent advancements and innovations in the field, covering topics ranging from the delivery of lipid nanoparticle-based formulations to the brain, the selective targeting capabilities of CPPs, and, notably, the synergistic effects of combining CPPs with nanoparticles. Lipid nanoparticle-based drugs have shown advantages in enhancing the solubility, stability, and bioavailability of therapeutic molecules in the treatment of neurodegenerative disorders. On the other hand, CPP-conjugated drugs have shown promise in improving cellular uptake and facilitating selective targeting. The development of dual delivery systems that harness the advantages of both nanoparticles and CPPs holds tremendous potential for the future of targeted delivery systems. Particular attention has been paid to the potential of PACAP peptide as a therapeutic agent in neuroprotective drug delivery approaches. Despite the inherent challenges of traversing the BBB, targeted drug delivery systems, including nanoparticulate carriers, dual drug-loaded lipid nanoparticles, and CPPs, have demonstrated their utility in enhancing BBB penetration. Furthermore, the concept of nose-to-brain drug delivery has emerged as a promising approach for delivering drugs to the brain. To fully comprehend the underlying mechanisms, mitigate potential toxicity concerns, and advance the development of novel targeted delivery systems, further studies will be necessary before translating this strategy into human clinical trials.

## Figures and Tables

**Figure 1 nanomaterials-13-03004-f001:**
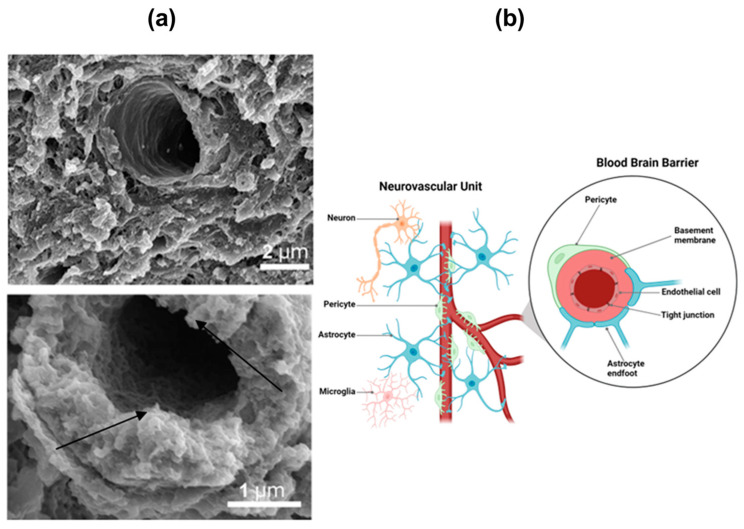
(**a**) Ultrastructure of the cerebral capillaries, constituting the regulatory interfaces between blood and brain (i.e., blood–brain barrier), revealed by scanning electron microscopy (SEM) at two magnifications (adapted with permission from [6]. Copyright © 2018, Nature Publishing Group, London, UK). The endothelial glycocalyx is present across the entire luminal surface of the cerebral capillaries (indicated by the black arrows). The dense glycocalyx structure enhances endothelial protection. (**b**) A schematic drawing of the cell types, which are present in the neurovascular unit that involves the BBB (created with BioRender). The neurovascular unit is formed by neurons, interneurons, astrocytic end-feet, microglia, oligodendrocytes, basal lamina covered with smooth muscular cells and pericytes, endothelial cells, and the extracellular matrix, as well as circulating blood components. Endothelial cells form the walls of the blood vessels. Astrocytes are the major glial cell type. Pericytes are cells of mesodermal origin embedded between the astrocyte end-feet and endothelium at the outer surface of the brain capillaries. Microglial cells are the resident immune cells of the CNS and interact with endothelial cells, astrocytes, and pericytes to regulate the BBB permeability and integrity [8,9,10].

**Figure 2 nanomaterials-13-03004-f002:**
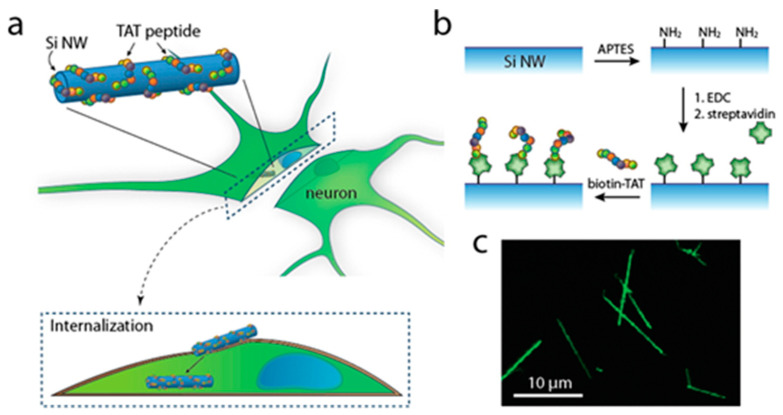
(**a**) Scheme of internalization of TAT peptide-modified Si nanowire (NW) into a neuron. (**b**) Conjugation scheme of a TAT peptide to the surface of Si NW. Streptavidin (STV) is first covalently attached via an EDC coupling reaction. The biotin–TAT peptide is then conjugated to STV. (**c**) Fluorescence microscope image of TAT-NWs labeled with STV–Alexa Fluor 555 conjugate (reproduced with permission from [104]; Copyright © 2016 American Chemical Society, Washington, DC, USA).

**Figure 3 nanomaterials-13-03004-f003:**
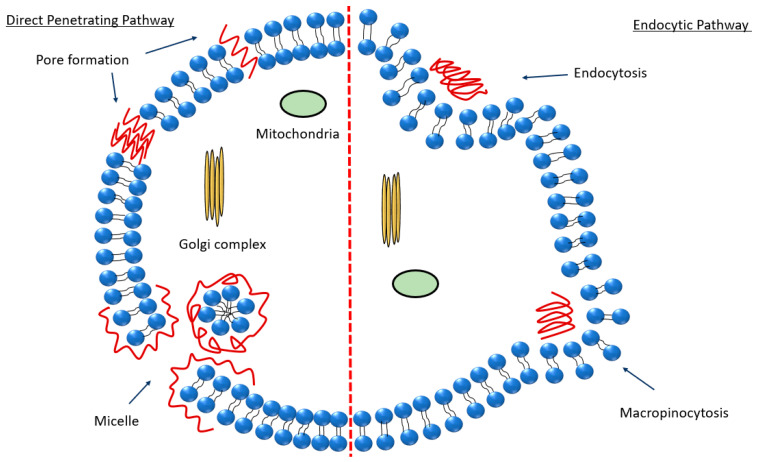
Scheme summarizing the pathways of peptide internalization discussed in the literature [124,125,126,127,128,129,130,131]. The endocytic pathway and the direct penetrating pathway are separated by a red dotted line.

**Figure 4 nanomaterials-13-03004-f004:**
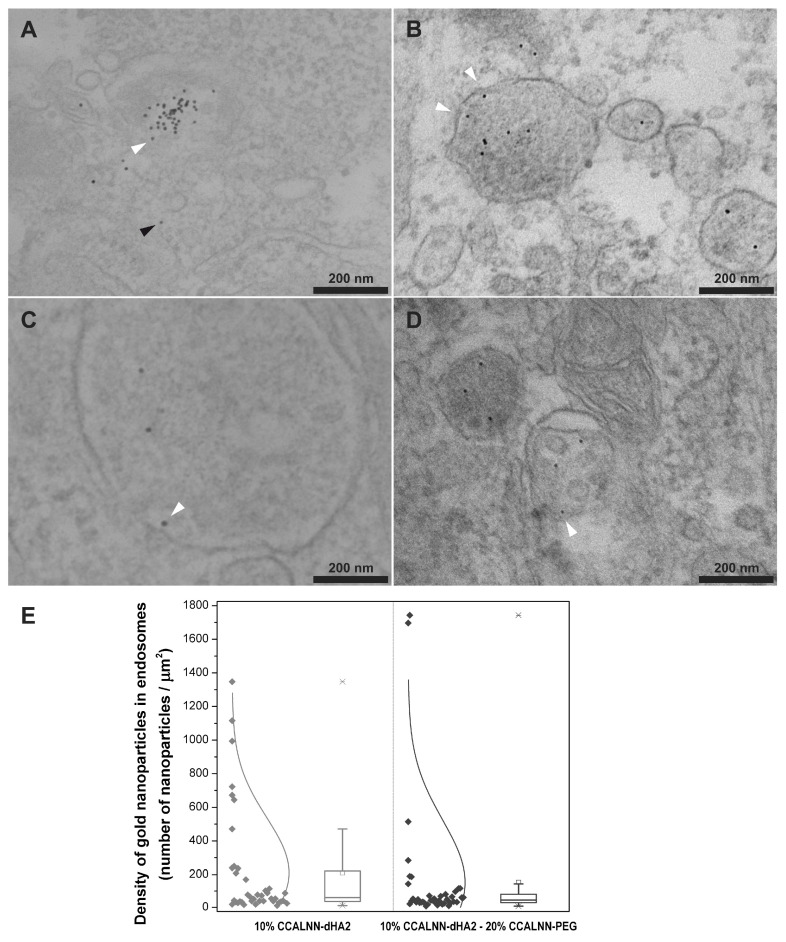
Cellular uptake and intracellular localization of HA2 fusion peptide-functionalized gold nanoparticles. (**A**,**B**) In the first scenario of the TEM images, the mixture consists of 10% CCALNN-dHA2 and 90% CALNN, while (**C**,**D**) in the second scenario, the composition includes 10% CCALNN-dHA2, 20% CCALNN-PEG, and 70% CALNN (Peptide sequences: CCALNN–HA2: CCALNNGGGGLFEAIEGFIENGWEGMIDGWYG: CCALNN–dHA2: CCALNNGdimGewGneifGaiaGflG-NH2; CCALNN-PEG: CCALNN-(ethylene glycol)6–glycinol, and CALNN). The arrowheads in the TEM images indicate the presence of gold nanoparticles, which can be observed either interacting with vesicular membranes (highlighted in white) or displaying a cytosolic distribution (highlighted in black). (**E**) The density of nanoparticles within endosomes was determined by analyzing the images and presented by light gray dots (for images with a sample size of *n* = 30) and in dark gray dots (for images (**C**,**D**) with a sample size of *n* = 29) [137]. Adapted with permission from [116] under Open Access Creative Commons Attribution License (*Plos.org*). The boxplots were constructed with edges spanning from the first to the third quartile. Whiskers extended 1.5 times the interquartile range from the box edges, while dashes denoted extreme values and crosses indicated the 1st and 99th percentiles. The distributions were analyzed using ten bins and included a fitted normal distribution curve.

**Figure 5 nanomaterials-13-03004-f005:**
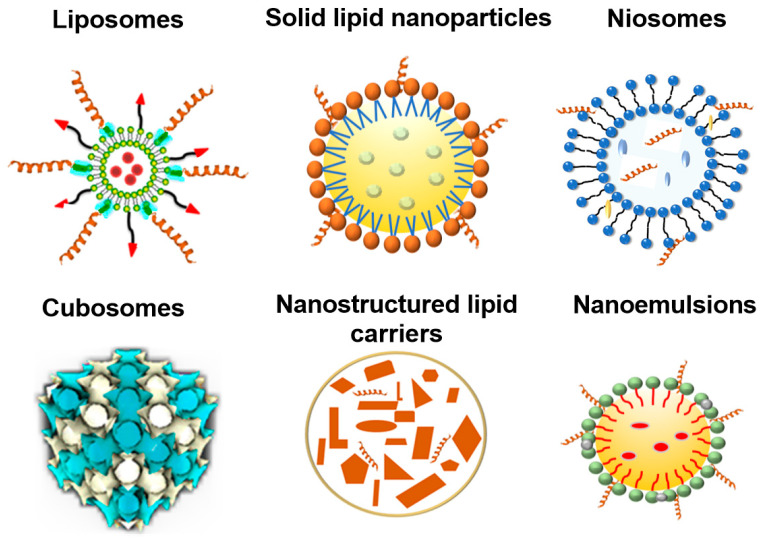
Variety of lipid nanoparticles, which can be functionalized by peptides for drug delivery in therapeutic strategies against neurological and other disorders. Examples of lipid-based nanocarriers include liposomes, solid lipid nanoparticles, niosomes, cubosomes, nanostructured lipid carriers, and nanoemulsions.

**Figure 6 nanomaterials-13-03004-f006:**
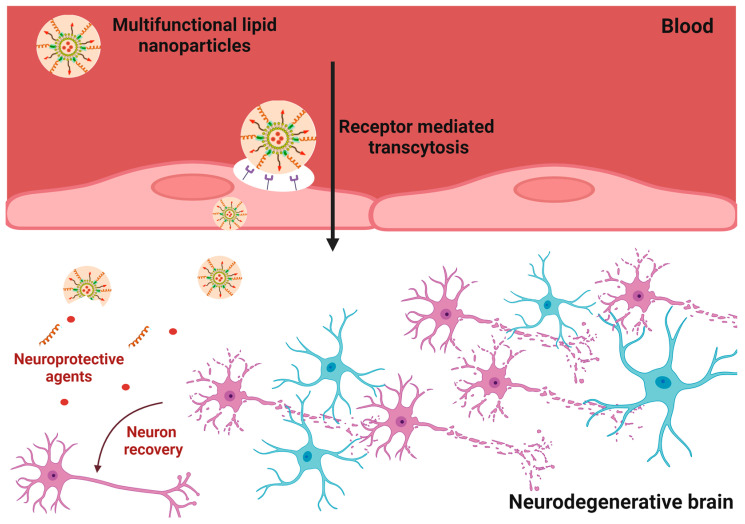
Pathways of nanoparticle internalization in nanotechnology approaches with neurodegeneration models (created with BioRender).

**Figure 7 nanomaterials-13-03004-f007:**
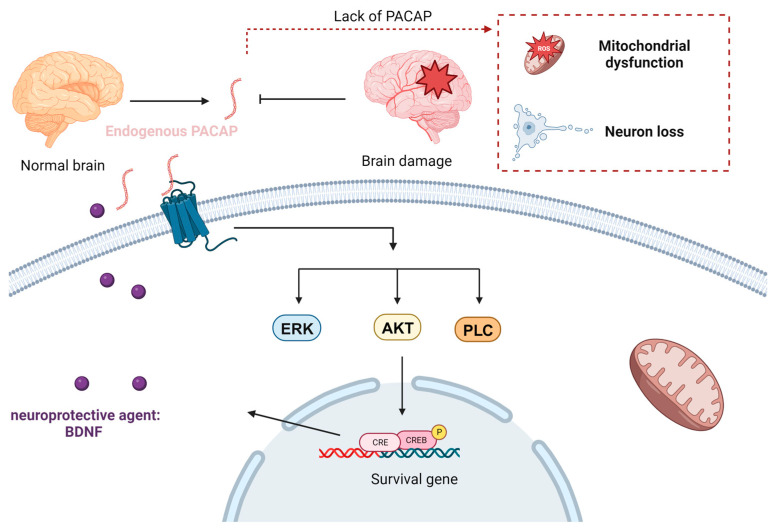
Scheme of PACAP functions, in which exogenous PACAP delivery promotes signaling pathways that support cellular survival. Conversely, the deficiency of PACAP leads to mitochondrial-mediated oxidative stress and neuronal apoptosis. This signaling cascade involves several key pathways, including the extracellular signal-regulated kinase (ERK) signaling pathway, also known as the ERK signaling pathway, the phosphatidylinositol 3-kinase-protein kinase B (PI3K-AKT) signaling pathway (often referred to as the AKT signaling pathway), and the phospholipase C (PLC) signaling pathway (created with BioRender).

**Figure 8 nanomaterials-13-03004-f008:**
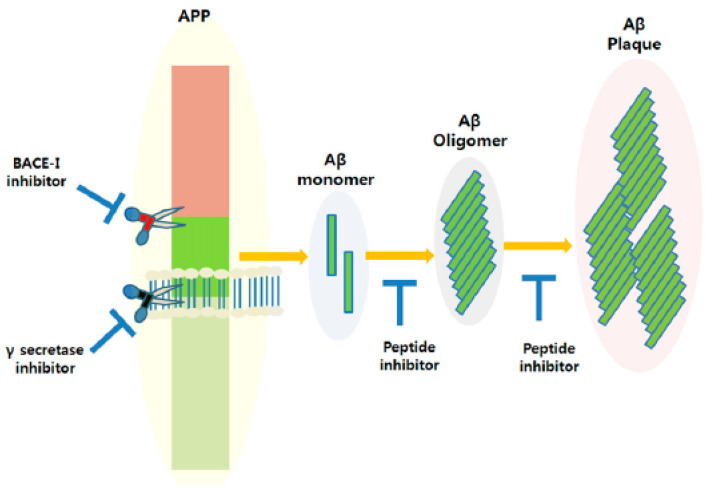
Scheme of using a peptide inhibitor as a peptide-based strategy for the treatment of Alzheimer’s disease (reprinted with permission from [187] under Open Access Creative Commons Attribution License (Frontiers Publisher, Lausanne, Suisse)).

**Figure 9 nanomaterials-13-03004-f009:**
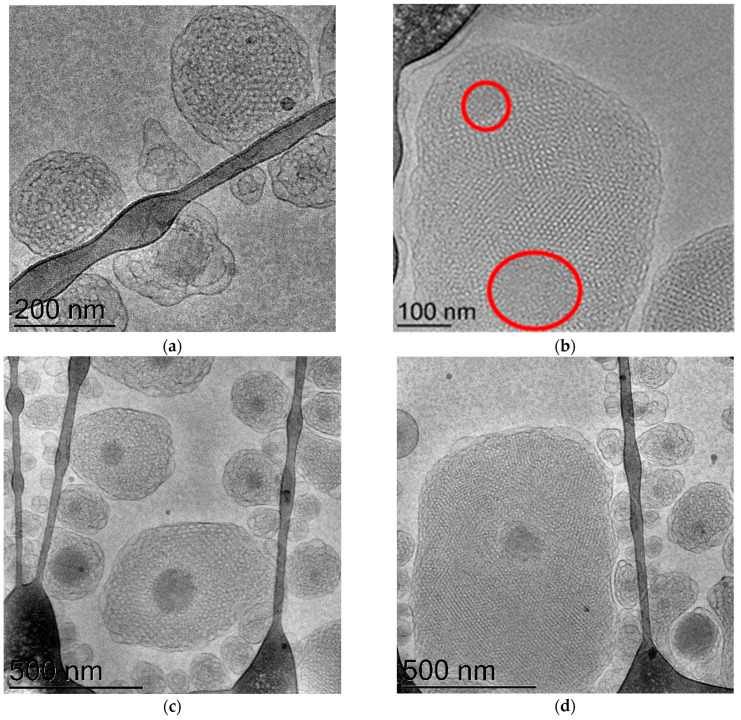
Cryo-TEM images of dispersed neuroprotective liquid crystalline lipid nanoparticles with encapsulated fish oil (rich in ω-3 polyunsaturated fatty acids (PUFAs)) and a phytochemical antioxidant [172]. (**a**) Cubosome and sponge type nanoparticles. The inner domains outlined by red color in panel (**b**) are rich in ω-3 PUFA oil, while the dark domains in the core of the cubosome liquid crystalline lipid nanoparticles correspond to the nanodomains of the encapsulated drug (**c**,**d**) (reproduced with permission from [99], Copyright © 2022 Elsevier, Amsterdam, The Netherlands).

**Figure 10 nanomaterials-13-03004-f010:**
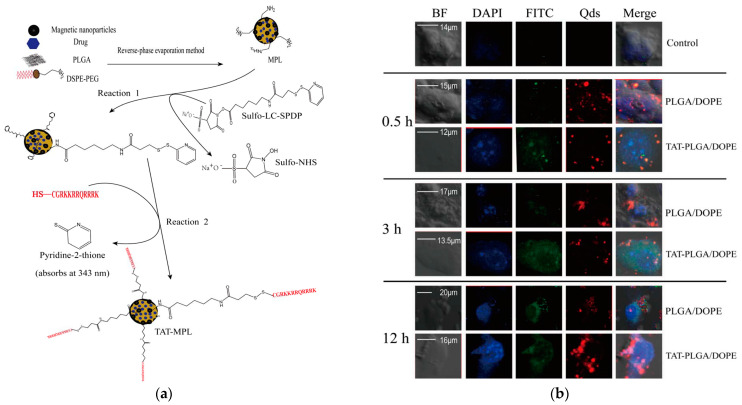
(**a**) Preparation scheme of trans-activating transcriptor (TAT) peptide-conjugated magnetic poly(D,L-lactide-co-glycolide) (PLGA)/lipid nanoparticles (MPLs) via conjugation with the TAT CPP sequence CGRKKRRQRRRK. The MPL nanoparticles combine the advantages of PLGA and magnetic liposomes for the encapsulation of the drugs hesperidin (HES), naringin (NAR), and glutathione (GSH). (**b**) Intracellular localization and distribution of QD-loaded fluorescein isothiocyanate (FITC)-labeled TAT-MPLs in bEnd.3 cells studied as an in vitro BBB model (reprinted with permission from the Public Library of Science, San Francisco, California, USA [105]).

**Table 2 nanomaterials-13-03004-t002:** Delivery approaches with various lipid-based nanosystems and their outcomes.

Lipid-Based System	Active Compound	Delivery Approach	Outcome	Ref.
Lipid liquid crystalline nanoparticles (Cubosomes)	Nerve growth factor (NGF)	Round window membrane administration (Guinea pigs)	Lipid cubosomes with encapsulated NGF enabled the overcoming of the barrier of the round window membrane (RWM) and enhanced the bioavailability of the NGF protein in the inner ear, with a promising potential for treating sensorineural hearing loss.	[158]
Lipid liquid crystalline nanoparticles (Cubosomes)	Model drug lissamine rhodamine (RhoB), a P-gp substrate and a molecule with poor BBB permeability	In vivo microinjection (Zebrafish larvae)	Cubosomes coated with Tween 80, Pluronic F127, or Pluronic F68 surfactants enabled the BBB targeting of the nanocarriers and enhanced the in vivo uptake of RhoB in Zebrafish.	[159]
Lipid liquid crystalline nanoparticles (Hexosomes)	Plasmalogen	In vitro (neuronal cell culture)	Lipid cubosomes and hexosomes, encapsulating plasmalogen, can significantly prolong the CREB activation up to 24 h.	[160]
TfR-targeting liposomes functionalized with different CPPs (TAT, pVec, QL)	Plasmid DNA	In vitro BBB model and intravenous administration (mice)	Biodistribution analysis revealed the enhanced targeting delivery of TAT-Tf liposomes in the brains of mice, with significantly increased fluorescent intensity.	[142]
TAT Lipid nanocapsules	D2-Glycerol ester (PGD2-G)	Intranasal administration (mice)	TAT-lipid nanocapsules were able to cross the olfactory monolayer and reach the CNS after nasal administration. TAT increased the portion of lipid-nanocapsules that reached the brain.	[161]
Gelatin nanostructured lipid carriers	Nerve growth factor (NGF)	Intravenous administration (rat)	NGF-gelatin nanostructured lipid carriers enhanced neuronal survival and contributed to improved functional recovery in a rat model of acute spinal cord injury.	[162]

## Data Availability

No data have been collected.

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
