# Peer review of "Recent Uses of Lipid Nanoparticles, Cell-Penetrating and Bioactive Peptides for the Development of Brain-Targeted Nanomedicines against Neurodegenerative Disorders"

_nanomaterials, 2023, doi:10.3390/nano13233004_

Round 1

Reviewer 1 Report

Comments and Suggestions for Authors

In recent years, there has been a growing body of literature focusing on the utilization of cell-penetrating peptides for enhancing drug delivery to the brain. Notably, the awarding of the Nobel Prize in Physiology or Medicine in 2023 to the creators of mRNA vaccines has significantly elevated interest in lipid nanoparticles (LNPs). Consequently, the interplay between LNPs and cell-penetrating peptides has emerged as a promising avenue for research in the field of brain-targeted drug delivery. Within this context, the PACAP peptide presents a novel and intriguing perspective, yet it remains an underexplored topic in review-type research. While the article exhibits a commendable depth of content, showcasing the author's evident familiarity with the latest advancements in the field, there are some key areas that warrant improvement. One noteworthy concern is the article's focus and logical flow, which could benefit from greater clarity. Some sections appear overly detailed, while essential aspects lack adequate explanation. For instance, the mention of neurodegenerative diseases in the abstract is somewhat disconnected from the article's title, leading to confusion about the study's precise objectives.

Given these concerns, I recommend a major revision of the article. The specific points of feedback are elaborated upon in greater detail below, and I kindly request that the author address each of these comments in their response.

Major

1. Abstract and Title: Ensure consistency between the abstract and title. Decide whether the focus is on neurodegenerative diseases, PACAP, or another specific aspect of your research. The title should reflect the main subject matter of the paper.

2. Introduction Relevance: Reduce the amount of information about central nervous system diseases in the introduction if it's not well integrated into the subsequent chapters. Focus on the most relevant information.

3. Clarify the relevance of mentioning macrophages on page 3, line 78. Ensure it connects logically with the previous and following content. If it's not relevant, consider removing it.

4. Include a brief description of hydrophobic peptides in Section 2.1, particularly if they are relevant to your research.

5. Improve the language when citing papers in Section 2.2 to make it more precise and clear.

6. Elaborate on the limitations of the peptides as mentioned on page 6, line 234. Explain why these limitations are important in the context of your research.

7. Clarify the significance of discussing different types of LNPs in Section 3.1. Explain how these different types are relevant to brain drug delivery.

8. Ensure that the literature cited in Page 12, line 451, supports the claim of the neuroprotective effect of PACAP.

9. Summarize the commonalities or relevance of the studies listed in Section 3.3, making it clear why they are mentioned.

10. Explain the connection between DNA vaccines and brain drug delivery when mentioning them on Page 13, line 494.

11. If Figure 7 is intended to illustrate the mechanism by which PACAP works, ensure that this is explained within the article's content.

12. Remove or rephrase statements in Page 15, line 558 that are not supported by the literature.

Minor:

Figure 1a may requires arrows indicating endothelial glycocalyx.

Page 3, line 110. The statement is not smooth.

Nose-to-brain drug delivery is indeed a good direction, but there is a lack of logic between the subtitles. Is there a better way to express it?

Comments on the Quality of English Language

No comments

Reviewer 2 Report

Comments and Suggestions for Authors

The article "Recent Uses of Lipid Nanoparticles and Cell Penetrating Peptides for Drug Delivery to the Brain: Emphasis on the Neurotrophic Therapeutic Peptide PACAP" effectively summarizes recent advancements and innovations in Lipid Nanoparticles and Cell Penetrating Peptides technology, focusing on their applications in the context of neurodegenerative diseases. The manuscript is well-organized and meticulously edited. However, I would like to offer some additional comments and questions:

The authors might consider emphasizing the current advantages and limitations of Lipid Nanoparticles and Cell Penetrating Peptides. Providing a discussion of strategies to address future opportunities and challenges in this field would enhance the manuscript's comprehensiveness.

I recommend minimizing the use of acronyms and abbreviations, especially when they appear only once in the text. This practice can help maintain the reader's focus and comprehension. For example:

Line 143: Instead of "LDL," use "Low-Density Lipoprotein."

Line 162: Replace "TAT" with "Trans-activator of Transcription."

Line 196: Spell out "MPG" as "Mannose-6-phosphate."

Line 211: Clarify "Tat-NTS" as "Trans-activator of Transcription-Neurotensin."

Line 230: Elaborate on "HAI" as "Hyaluronic Acid-Modified."

Line 279: Define "VPAC1" as "Vasoactive Intestinal Peptide Receptor 1."

Lines 349 and 350: Explain "SH-SY5Y" as "Human Neuroblastoma Cells" and "CAT" as "Catalase."

Lines 436, 443, 456, 461, and 477: Expand "NR2B" to "N-Methyl-D-Aspartate Receptor Subunit 2B," "cAMP" to "Cyclic Adenosine Monophosphate," "GFP" to "Green Fluorescent Protein," "HR9" to "Histone Replacement 9," and "KLA" to "Kappalactone A.

Comments on the Quality of English Language

No comment.

Reviewer 3 Report

Comments and Suggestions for Authors

It was a pleasure to review Wu and Angelova‘s manuscript titled „Recent uses of lipid nanoparticles and cell penetrating peptides for drug delivery to the brain: emphasis on the neurotrophic therapeutic peptide PACAP“. Angelina Angelova is well-known for her research in the field of lipid-based nano-assemblies. However, I find the review too broad, superficial, and unrelated to the title. I would expect to learn more about the potential benefits of this combined strategy of using lipid nanoparticles and cell-penetrating peptides (CPPs) to cross the brain-blood barrier.

The Abstract and Introduction sections are written quite well. However, the following list of issues in other sections of this manuscript:

1.      The application of lipid-based structures in the treatment of neurological diseases has already been documented in a review paper published in Pharmaceutics by Witika et al. (2022) titled “Lipid-based nanocarriers for neurological disorders: a review of the state-of the-art and therapeutic success to date“. Thus, I would expect the current submitted review to go into greater detail regarding the selected topic, including the advantages and disadvantages, of combining such lipid structures with cell-penetrating peptides. However, it is not a well-organized story with too much information and no real significance. Thus, this paper should critically evaluate why these systems still have trouble entering the market.

2.      I would expect the description of a detailed mechanism on how lipid-based drug vehicles (LBDV) overcome the brain-blood barrier. How would CPPs overcome this barrier, where they struggle, what is the mechanism, and where would the co-usage of CPP and LBDV benefit?

3.      Despite this, as I mentioned, the manuscript is too broad and sometimes unrelated to the subject. Example, Section 3.2. Do nanoparticles cross the blood-brain barrier? – this either should be expanded or removed. The authors are questioning do nanoparticles cross the barrier, but they are comparing nanoparticles of different physical properties (polymer, metal, lipid-based) with no additional discussion on the size of these particles. Thus, I don‘t see any value in describing other systems than lipid-based ones.

4.      Some minor comments: authors overuse self-citations despite at least several groups worldwide working on the same topic (country of origin, Australia or Sweden, etc.). The authors are focusing on the limited number of lipid-based drug vehicles. Why hexosomes or sponge-like nanoparticles are omitted?

5.      Is the combination of CPPs and lipid nanoparticles proof of concept? The manuscript lacks to highlight this.

6.      Using visuals is always a nice touch to explain the written word, but here, I see the Figures as not adds-on but separate inserts. Figure 10 – why the reaction scheme is presented but not described in the text. Does it have value?

Overall, the manuscript is well-written but should focus more on the premises described in the title. The main concern is that this review is more descriptive, where I expected more dialogue on how to make that chosen system work.

Reviewer 4 Report

Comments and Suggestions for Authors

The manuscript entitled, ‘Recent Uses of Lipid Nanoparticles and Cell Penetrating Peptides for Drug Delivery to the Brain: Emphasis on the Neurotrophic Therapeutic Peptide PACAP’ reported PE based coated fabric for EMI shielding applications. The article should be modified according to the following points;

1.      First of all the abstract section is too large. Better to concise it.

2.      Why DD is important by lipid based systems compared to others should be emphasized also with some references.

3.      Modify this heading which is closely related to the topic.

4.      Some table would be helpful to compare this lip based systems in different types of delivery approaches.

5.      Some articles have significance (related to sensing) and could be discussed with the help of following references:

(a)    Ganguly, S., & Margel, S. (2021). Design of magnetic hydrogels for hyperthermia and drug delivery. Polymers13(23), 4259.

(b)   Maity, P. P., Kapat, K., Poddar, P., Bora, H., Das, C. K., Das, P., ... & Dhara, S. (2023). Capra cartilage-derived peptide delivery via carbon nano-dots for cartilage regeneration. Frontiers in Bioengineering and Biotechnology11.

(c)    Liu, R., Luo, C., Pang, Z., Zhang, J., Ruan, S., Wu, M., ... & Gao, H. (2023). Advances of nanoparticles as drug delivery systems for disease diagnosis and treatment. Chinese chemical letters34(2), 107518.

Round 2

Reviewer 1 Report

Comments and Suggestions for Authors

The authors have answered all my questions.

Reviewer 3 Report

Comments and Suggestions for Authors

Thank you for your answers to my comments. The idea behind is more clear and persuasive.

Reviewer 4 Report

Comments and Suggestions for Authors

This can be published in its present form.